# LOCAL EXPLANATION METHODS FOR DEEP NEURAL NETWORKS LACK SENSITIVITY TO PARAMETER VALUES

**Julius Adebayo, Justin Gilmer, Ian Goodfellow, & Been Kim**
Google Brain

## ABSTRACT

Explaining the output of a complicated machine learning model like a deep neural network (DNN) is a central challenge in machine learning. Several proposed local explanation methods address this issue by identifying what dimensions of a single input are most responsible for a DNN's output. The goal of this work is to assess the sensitivity of local explanations to DNN parameter values. Somewhat surprisingly, we find that *DNNs with randomly-initialized weights produce explanations that are both visually and quantitatively similar to those produced by DNNs with learned weights.* Our conjecture is that this phenomenon occurs because these explanations are dominated by the lower level features of a DNN, and that a DNN's architecture provides a strong prior which significantly affects the representations learned at these lower layers. **NOTE: This work is now subsumed by our recent manuscript, Sanity Checks for Saliency Maps (to appear NIPS 2018), where we expand on findings and address concerns raised in Sundararajan & Taly (2018).**

## 1 INTRODUCTION

Understanding how a trained model derives its output, as well as the factors responsible, is a central challenge in machine learning (Vellido et al., 2012; Doshi-Velez et al., 2017). A local explanation identifies what dimensions of a single input was most responsible for a DNN's output. As DNNs get deployed in areas like medical diagnosis (Rajkomar et al., 2018) and imaging (Lakhani & Sundaram, 2017), reliance on explanations has grown. Given increasing reliance on local model explanations for decision making, it is important to assess explanation quality, and characterize their fidelity to the model being explained (Doshi-Velez & Kim, 2017; Weller, 2017). Towards this end, *we seek to assess the fidelity of local explanations to the parameter settings of a DNN model*. We use random initialization of the layers of a DNN to help assess parameter sensitivity.

**Main Contributions**

- We empirically assess local explanations for faithfulness by re-initializing the weights of the models in different ways. We then measure the similarity of local explanations for DNNs with random weights and those with learned weights, and find that these sets of explanations are both visually and quantitatively similar.

- With evidence from prior work (Ulyanov et al., 2017; Saxe et al., 2011), we posit that these local explanations are mostly invariant to random initializations because they capture low level features that are mostly dominated by the input.

- Specifically, we hyptothesize that dependence of local explanations on lower level features is as follows: if we decompose the function learned by a DNN as $f(g(x; \gamma); \theta)$, where $g(x; \gamma)$ corresponds to the function learned by the lower layers, and $f(.; \theta)$ corresponds to the upper layers, then a local explanation, $E(x_t)$, for input $x_t$ corresponds to $E(x_t) \propto h(g(x; \gamma))$, where $h$ captures the intricacies of the local explanation methodology.

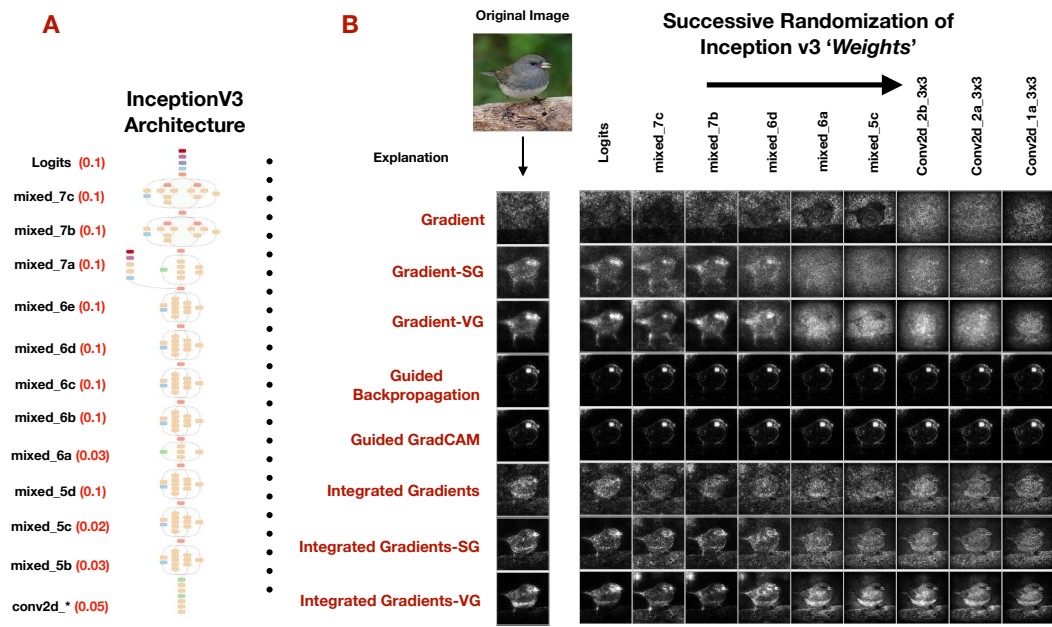

Figure 1: **Change in explanations for various methods as each successive inception block is randomized, starting from the logits layer.** **A**: Inception v3 architecture along with the names of the different blocks. The number in the parenthesis is the top-1 accuracy of the Inception model on a test set of 1000 images after randomization up that block. Initial top-1 accuracy for this class of images was 97 percent. Conv2d* refers collectively to the last 5 convolutional layers. **B-Left**: Shows the original explanations for the Junco bird in the first column as well as the label for each explanation type shown. **B-Right:** Shows successive explanations as each block is randomized. We show images for 9 blocks of randomization. Coordinate (Gradient, mixed7b) shows the gradient explanation for the network in which the top layers starting from Logits up to mixed7b have been reinitialized. The last column corresponds to a network where all the weights have been completely reinitialized. See Appendix for more examples.

## 2 LOCAL EXPLANATION METHODS

In this section, we provide an overview of the explanation method examined in this work. In our formal setup, an *input* is a vector $x \in \mathbb{R}^d$. A *model* describes a function $S : \mathbb{R}^d \to \mathbb{R}^C$, where $C$ is the number of *classes* in the classification problem. An explanation method provides an *explanation map* $E : \mathbb{R}^d \to \mathbb{R}^d$ that maps inputs to objects of the same shape.

### 2.1 OVERVIEW OF METHODS

The **gradient explanation** for an input $x$ is $E_{\text{grad}}(x) = \frac{\partial S}{\partial x}$ (Baehrens et al., 2010; Simonyan et al., 2013). The gradient quantifies how much a change in each input dimension would a change the predictions $S(x)$ in a small neighborhood around the input.

**Integrated Gradients (IG)** also addresses gradient saturation by summing over scaled versions of the input (Sundararajan et al., 2017). IG for an input $x$ is defined as $E_{\text{IG}}(x) = (x - \bar{x}) \times \int_0^1 \frac{\partial S(\bar{x} + \alpha(x - \bar{x}))}{\partial x} d\alpha$, where $\bar{x}$ is a "baseline input" that represents the absence of a feature in the original input $x$.

**Guided Backpropagation (GBP)** (Springenberg et al., 2014) builds on the "DeConvNet" explanation method Zeiler & Fergus (2014) and corresponds to the gradient explanation where negative gradient entries are set to zero while back-propagating through a ReLU unit.

**Guided GradCAM.** Introduced by Selvaraju et al. (2016), GradCAM explanations correspond to the gradient of the class score (logit) with respect to the feature map of the last convolutional unit

of a DNN. For pixel level granularity GradCAM, can be combined with Guided Backpropagation through an element-wise product.

**SmoothGrad (SG)** (Smilkov et al., 2017) seeks to alleviate noise and visual diffusion (Sundararajan et al., 2017; Shrikumar et al., 2016) for saliency maps by averaging over explanations of noisy copies of an input. For a given explanation map $E$, SmoothGrad is defined as $E_{\text{sg}}(x) = \frac{1}{N}\sum_{i=1}^{N} E(x+g_i)$, where noise vectors $g_i \sim \mathcal{N}(0,\sigma^2))$ are drawn i.i.d. from a normal distribution.

**VarGrad (VG)** is a variance analog to SmoothGrad. VarGrad is defined as: $E_{\text{vg}}(x) = \mathcal{V}(E(x+g_i))$ where noise vectors $g_i \sim \mathcal{N}(0,\sigma^2))$ are drawn i.i.d. from a normal distribution, and $\mathcal{V}$ corresponds to the variance.

**Summary.** The local explanation methods described so far are the ones considered in this work. Our selection criteria for the approaches included for analysis was based on ease of implementation, running time, and memory requirements. While there are several different approaches for obtaining local-explanations (Ribeiro et al., 2016; Fong & Vedaldi, 2017; Dabkowski & Gal, 2017; Zintgraf et al., 2017; Pieter-Jan Kindermans, 2018), recent work by Ancona et al. (2017) and Lundberg & Lee (2017) both show that there are equivalences among several of the previously proposed methods.

# 3   LOCAL EXPLANATION SPECIFICITY

In this section, we provide a discussion of the key results from out experiments. See attached Appendix for discussion of experimental details and additional figures demonstrating out results. First, we randomly reinitialize the weights of a DNN starting from the top layers going all the way to the first layer. Second, we independently reinitialize the weights of each layer. With these randomization schemes, we then visually, and quantitatively assess the change in local explanations for a model with learned weights and one with randomized weights. To quantitatively assess the similarity between two explanations, for a given sample, we use the Spearman's rank order correlation metric inspired by the work of Ghorbani et al. (2017).

**Cascading Network Randomization - Inception v3 on ImageNet.** As figure 3 indicates, guided back-propagation and guided grad-CAM show no change in the explanation produced regardless of the degradation to the network. We observe an initial decline in rank correlation for integrated gradients and gradients, however, we see a remarkable consistency as well through major degradation of the network, particularly the middle blocks. Surprisingly, the input-output gradient shows the most change of the methods tested as the re-initialization approaches the lower layers of the network. We observe similar results for a CNN and MLP on MNIST.

**The architecture is a strong prior.** Saxe et al. (2011) showed that features learned from CNNs with random weights perform surprisingly well in a downstream classification task when fed to a linear classifier. The authors showed that certain CNNs with random weights still maintain translation in-variance and are frequency selective. To further this point, Alain & Bengio (2016) find that the features extracted from a randomly-initialized 3-hidden layer CNN on MNIST (the same architecture that is used in this work) lead to a 2 percent test error accuracy. As part of their analysis, they find that the best performing randomly-initialized features are those derived right after the Relu activation of features derived from the first layer. Taken together, these findings highlight the following:

*the architecture of DNN is a strong prior on the input, and with random initialization, is able to capture low-level input structure particularly for images.*

# 4   CONCLUSION

We empirically assess local explanations of a DNN derived from several different methods to find that DNNs with randomly-initialized weights produce explanations that are both visually and quantitatively similar to those produced by DNNs with learned weights. We posit that this phenomenon occurs because local explanations are dominated by lower level features, and that a DNN's architecture provides a strong prior which significantly affects the representations learned at these lower layers even for randomly-initialized weights.

ACKNOWLEDGMENTS

We thank Jaime Smith for providing the idea for VarGrad. We thank members of the Google PAIR team for open-source implementations of the local explanation methods used in this work and for providing useful feedback.

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

## SUCCESSIVE RANDOMIZATION VISUALIZATION

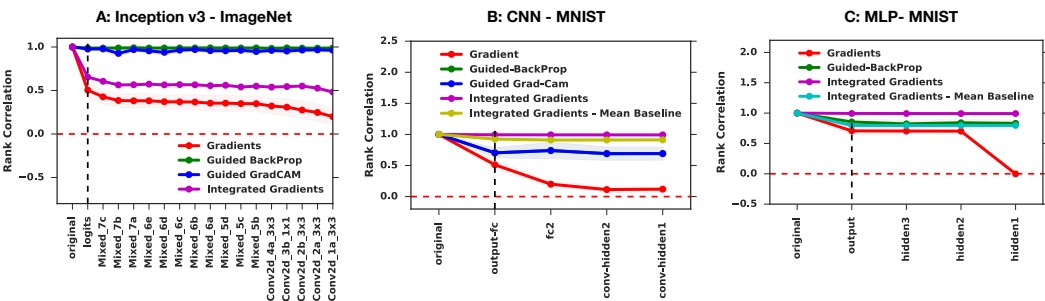

Figure 2: **Successive reinitialization starting from top layers for (A) Inception v3 on ImageNet, (B) CNN on MNIST, and (C) MLP on MNIST.** In all plots, y axis is the rank correlation between original explanation and the randomized explanation derived for randomization up to that layer or block (inception), while the x axis corresponds to the layers/blocks of the DNN starting from the output layer. The black dashed line indicates where successive randomization of the network begins, which is at the first layer. A: Rank correlation plot for Inception v3 trained on ImageNet. B: Rank correlation explanation similarity plot for a 3 hidden-layer CNN on MNIST. C: Rank correlation plot for a 3-hidden layer feed forward network on MNIST.

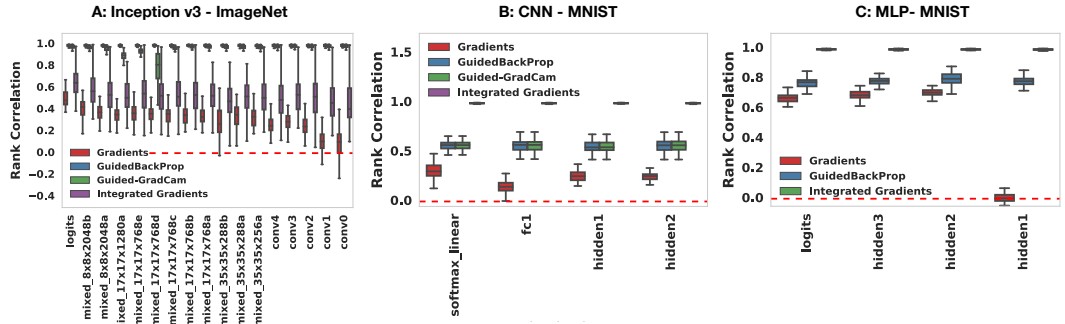

Figure 3: **Independent reinitialization of each layer (A) Inception v3 on ImageNet, (B) CNN on MNIST, and (C) MLP on MNIST.** In all plots, y axis is the rank correlation between original explanation and the randomized explanation derived for independent randomization of that layer or block (inception), while the x axis corresponds to the layers/blocks of the DNN starting from the output layer. The red dashed line indicates zero rank correlation. A: Rank correlation plot for InceptionV3 trained on ImageNet. B: Rank correlation explanation similarity plot for a 3 hidden-layer CNN on MNIST. C: Rank correlation plot for a 3-hidden layer feed forward network on MNIST.

## BACKGROUND & EXPERIMENTAL DETAILS

We give an overview of the models and data sets used in our experiments.

**Data sets.** To perform our randomization tests, we used the following data sets: ILSVRC 2012 (ImageNet classification challenge data set) Russakovsky et al. (2015), and the MNIST data set LeCun (1998).

**Models.** we perform our randomization tests on a variety of models across the data sets previously mentioned as follows: a pre-trained Inception v3 model Szegedy et al. (2016) on ImageNet dataset, a multi-layer perceptron model (3 hidden layers) trained on the MNIST, a 3 hidden layer CNN also trained on the MNIST.

**Explanation Similarity.** To quantitatively assess the similarity between two explanations, for a given sample, we use the Spearman's rank order correlation metric inspired by Ghorbani et al. (2017). The key utility of a local explanation lies in its ranking of the relevant of parts of the input; hence, a natural metric for comparing the similarity in conveyed information for two local explanations is the Spearman rank correlation coefficient between two different explanations. For a quantitative comparison on each dataset, we use a test bed of 200 images. For example, for the ImageNet dataset, we selected 200 images from the validation set that span 10 highly varied classes ranging from Banjo to Scuba diving. For MNIST, we randomly select 200 images from the test data on which to compute explanations for each of the tested methods. In the figures shown indicating rank correlation between true and randomized models, each point is an average of the correlation coefficient over 200 images, and the 1-std band is shown around each curve to provide a sense for the variability of the rank correlation estimate.

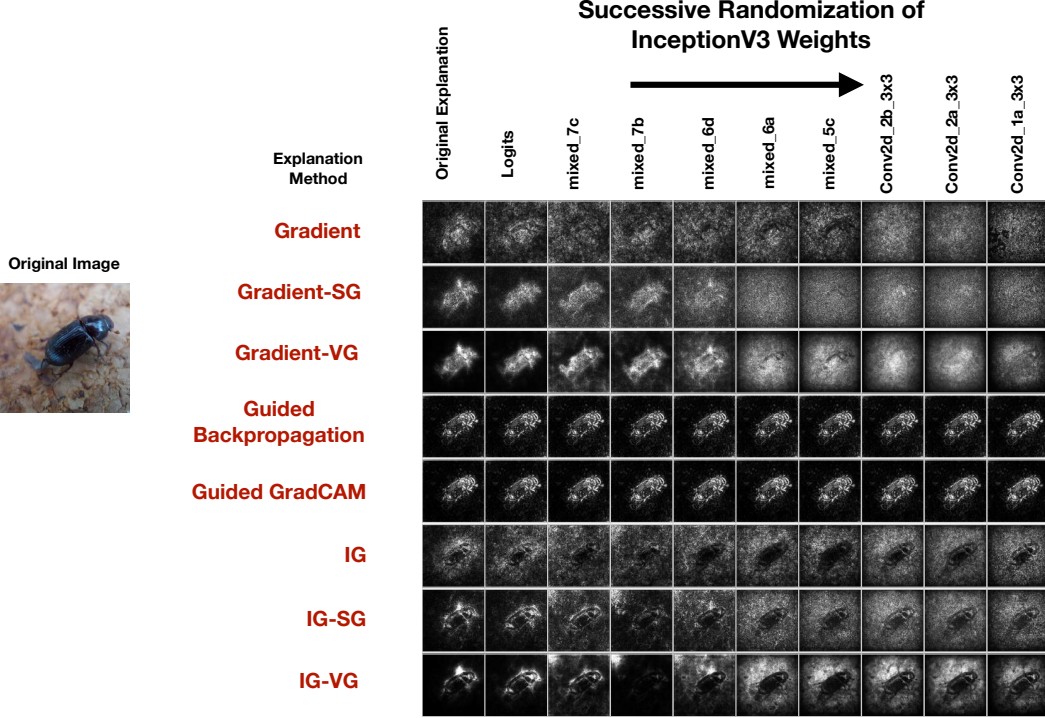

Figure 4: **Successive Randomization Bug.**

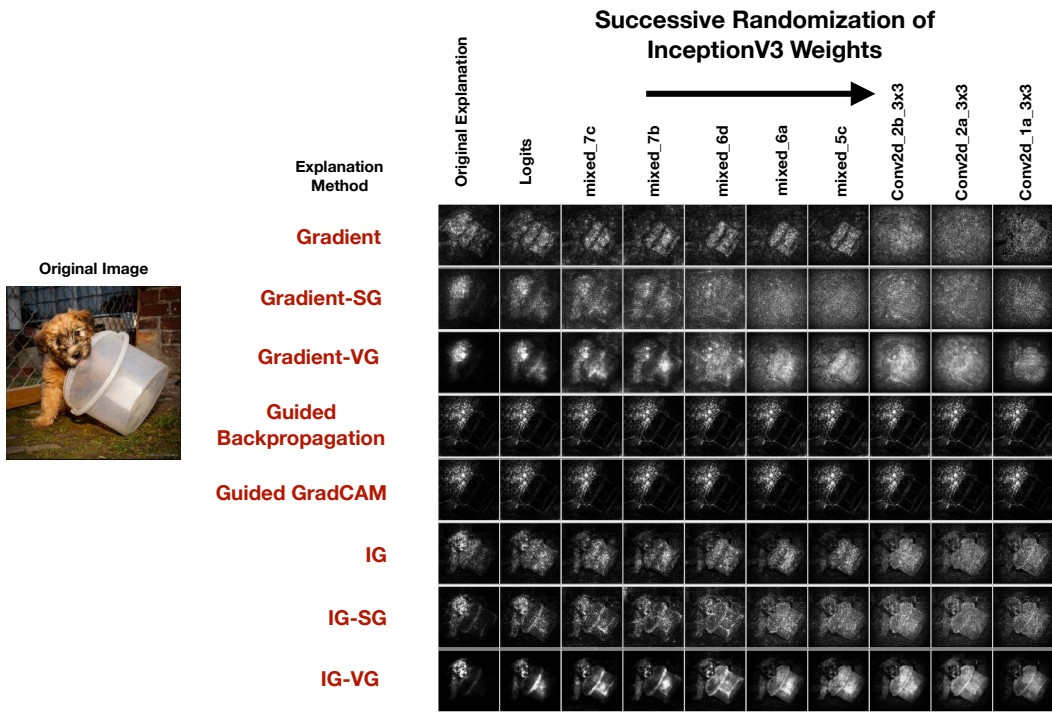

Figure 5: **Successive Randomization Dog with bucket.**

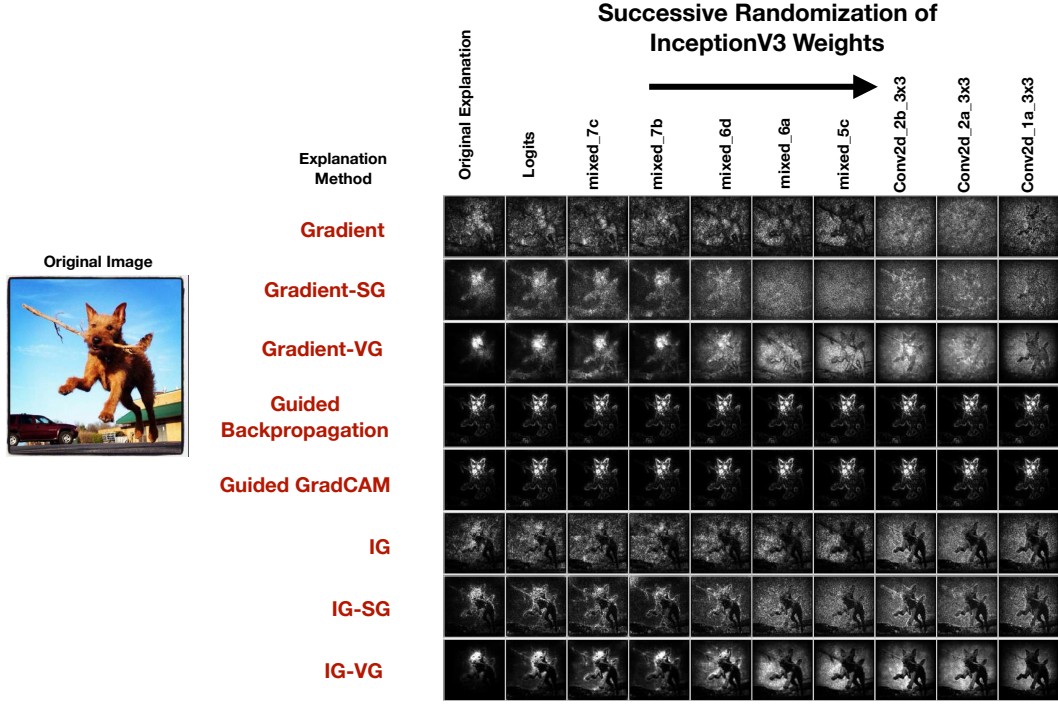

Figure 6: **Successive Randomization Dog with Stick.**

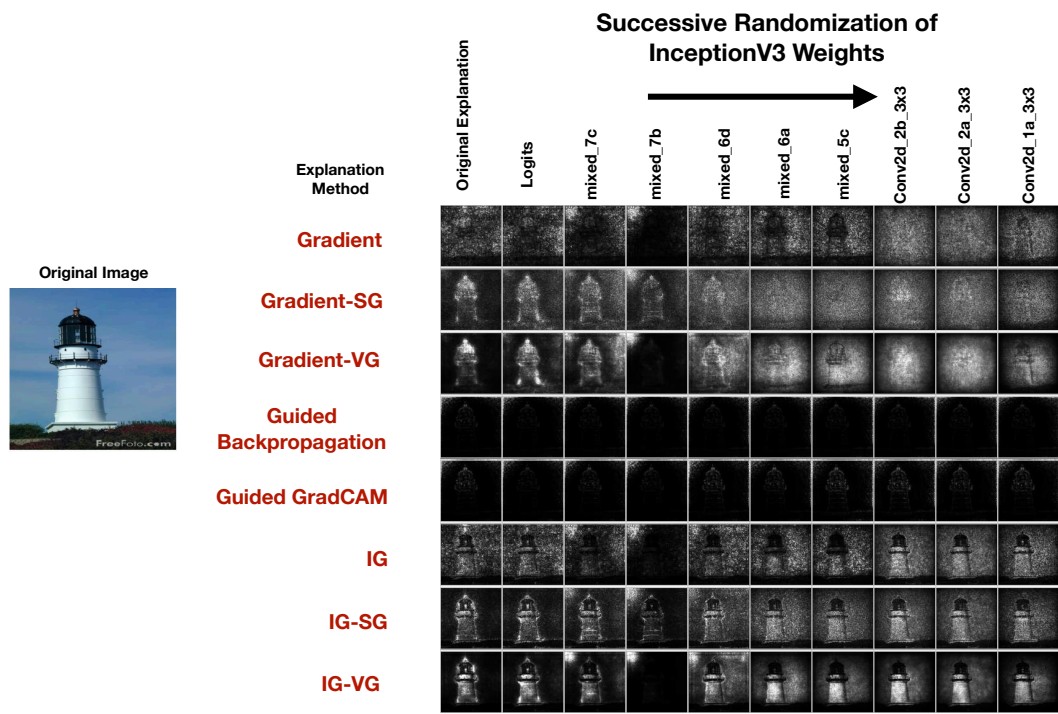

Figure 7: **Successive Randomization Light House.**

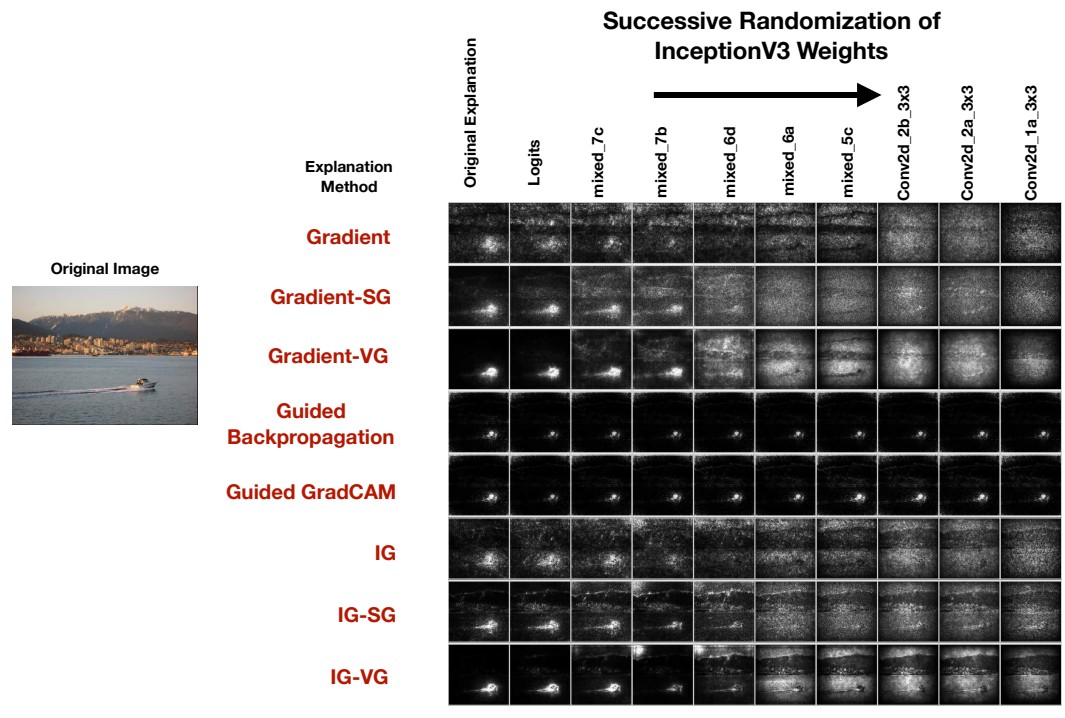

Figure 8: **Successive Randomization Speedboat.**

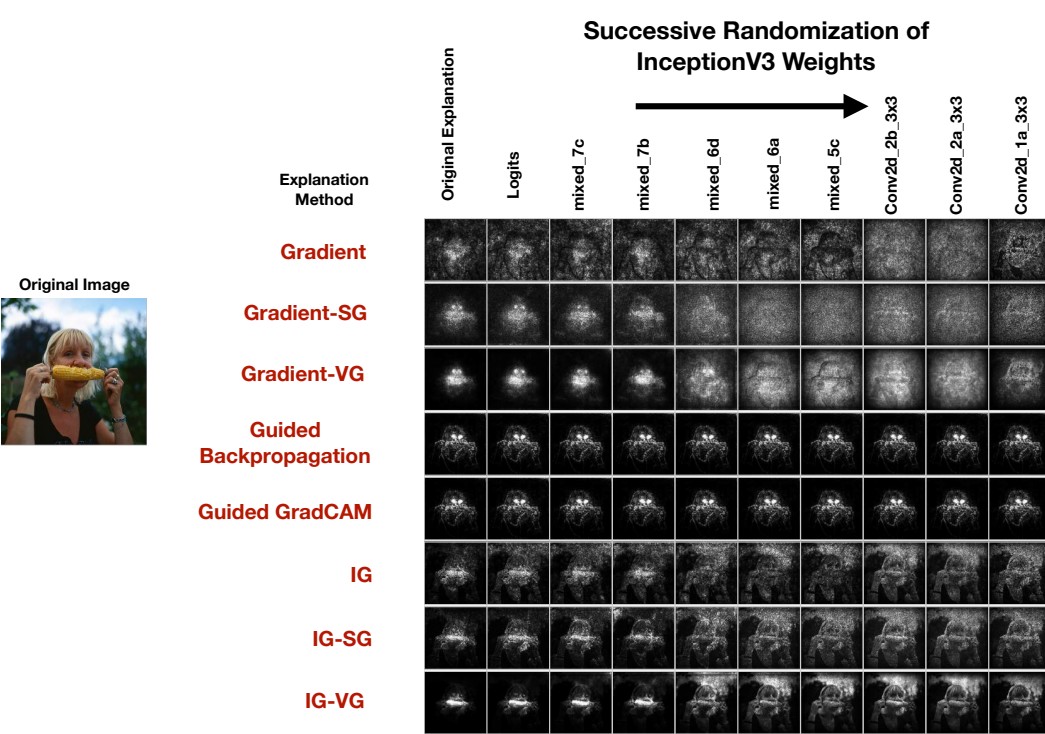

Figure 9: **Successive Randomization Corn.**

