# OpenReview forum: "Local Explanation Methods for Deep Neural Networks Lack Sensitivity to Parameter Values"
_ICLR.cc/2018/Workshop — Accept_

### Official Review · AnonReviewer3 · 2018-03-09
**very insightful paper**

**Rating:** 9
**Confidence:** 4

**Review:**

this paper is very well written and insightful.

it made an attempt to address model interpretability in deep learning. the authors provide very curious and interesting results suggesting the interaction between the network architecture and interpretability of the weights via local explanation. I believe this paper would be an asset to the iclr workshop and could spawn many further explorations.

---

### Official Review · AnonReviewer2 · 2018-03-10
**Work provides only very limited insights**

**Rating:** 4
**Confidence:** 5

**Review:**

This paper empirically assesses the stability of few (mainly gradient-based) local explanation methods. Results show that explanations are not much affected by random reinitialization of higher layers of the DNN.

Although the stability of explanation method is an important research topic, this work provides only very limited insights into the problem.
1) The lack of sensitivity is empirically measured, but an in-depth analysis of the results is lacking. For instance, it remains unclear if the lack of sensitivity is due to the structure of the DNN model or the imperfection of the evaluated explanation methods. It also remains unclear why the Gradients show a slightly different behavior than the other methods.

2) The authors ignore an important class of explanation methods which decompose the classification function, e.g., LRP (Bach et al. 2015), Excitation Backprop (Zhang et al. 2016), Deep Taylor Decomposition (Montavon et al. 2017). These three closely related methods have advantages in terms of explanation continuity and explanation selectivity over gradient-based approaches (see discussion in Montavon et al. 2018) and thus may show very different behaviour in randomization experiments. Actually, Zhang et al. demonstrated that Excitation Backprop (in contrast to other explanation methods) produces discriminative explanations. Thus, for Excitation Backprop (similare results are expected for alpha-beta LRP and Deep Taylor Decomposition) the signal at higher layers "does matter". Therefore, I believe that these three explanation methods will be sensitivity to parameters values and will show very different behaviour in randomization experiments.

---

### Official Review · AnonReviewer1 · 2018-03-10
**good paper**

**Rating:** 7
**Confidence:** 4

**Review:**

This paper shows that local explanations for DNNs with random-initialized weights are qualitatively and quantitatively similar to explanations produced by DNNs with learned weights.

Pros:
The paper is clear, the problem is well stated and the method is sound.

Cons:
The impact of the findings in this paper is unclear. Perhaps the most important point made in the paper is the importance of the architecture over fine-tuning of the weights for explanation tasks (and more in general).

---

### Decision · Program_Chairs · 2018-03-20
**ICLR 2018 Workshop Acceptance Decision**

**Decision:**

Accept

**Comment:**

Congratulations, your paper was accepted to the ICLR workshop.